# Dietary Oregano Essential Oil Supplementation Influences Production Performance and Gut Microbiota in Late-Phase Laying Hens Fed Wheat-Based Diets

**DOI:** 10.3390/ani12213007

**Published:** 2022-11-02

**Authors:** Fei Gao, Lianhua Zhang, Hui Li, Fei Xia, Hongtong Bai, Xiangshu Piao, Zhiying Sun, Hongxia Cui, Lei Shi

**Affiliations:** 1Key Laboratory of Plant Resources and Beijing Botanical Garden, Institute of Botany, Chinese Academy of Sciences, Beijing 100093, China; 2China National Botanical Garden, Beijing 100093, China; 3College of Pharmacy, Shandong University of Traditional Chinese Medicine, Jinan 250355, China; 4State Key Laboratory of Animal Nutrition, College of Animal Science and Technology, China Agricultural University, Beijing 100193, China

**Keywords:** essential oils, production performance, fatty acid composition, bacterial community, laying hen

## Abstract

**Simple Summary:**

Essential oils (**EOs**) are antibacterial and anti-inflammatory, and are called natural antibiotics. Within the great variety of EOs, oregano essential oil(OEO) is known for its antimicrobial activity. So, we focused on comparing the effect of flavomycin and OEO on the egg-laying performance and intestinal flora of laying hens. This study found that OEO improved the egg-production performance and altered microbial composition. The results revealed that OEO could be an effective alternative to flavomycin.

**Abstract:**

This study aimed to investigate the potential effects of OEO on production performance, egg quality, fatty acid composition in yolk, and cecum microbiota of hens in the late phase of production. A total of 350 58-week-old Jing Tint Six laying hens were randomly divided into five groups: (1) fed a basal diet (control); (2) fed a basal diet + 5 mg/kg flavomycin (AGP); (3) fed a basal diet + 100 mg/kg oregano essential oil + 20 mg/kg cinnamaldehyde (EO1); (4) fed a basal diet + 200 mg/kg oregano essential oil + 20 mg/kg cinnamaldehyde (EO2); (5) fed a basal diet + 300 mg/kg oregano essential oil + 20 mg/kg cinnamaldehyde (EO3). Compared to the control group, group EO2 exhibited higher (*p* < 0.05) egg production during weeks 5–8 and 1–8. EO2 had a lower feed conversion ratio than the control group during weeks 1–8. The content of monounsaturated fatty acid (MUFA) in EO2 was higher (*p* < 0.05) than that of the control and AGP groups. EO2 increased (*p* < 0.05) the abundance of Actinobacteriota and decreased the abundance of *Desulfovibri* in the cecum. The abundances of *Anaerofilum*, *Fournierella*, *Fusobacterium*, and *Sutterella* were positively correlated with egg production, feed conversion ratio, and average daily feed intake, while the abundances of *Bacteroides*, *Desulfovibrio*, *Lactobacillus*, *Methanobrevibacter*, and *Rikenellaceae_RC9_gut_group* were negatively correlated with egg production, feed conversion ratio, and average daily feed intake. Dietary supplementation with 200 mg/kg OEO and 20 mg/kg cinnamaldehyde could improve egg-production performance, decrease feed conversion ratio, and alter the fatty acid and microbial composition of eggs from late-phase laying hens.

## 1. Introduction

Antibiotics have been widely used in the poultry industry as animal-growth promoters for decades. However, the long-term use of sub-therapeutic antibiotics caused a series of problems, such as making bacteria resistant to antibiotics, antibiotic residues, endogenous infections, and double infections, which pose serious threats to livestock and poultry production and human health [1,2]. Now, China has banned feed antibiotics, leading to increased demand for green feed additives.

Essential oils (EOs) are concentrated hydrophobic liquids containing volatile aromatic compounds obtained from plants [3]. Oregano essential oil (OEO) has strong antibacterial properties on various pathogenic bacteria, such as *Salmonella enteritidis*, *Escherichia coli*, *Staphylococcus aureus*, *Campylobacter*, etc. [4,5,6], and is considered to be a substitute for antibiotics. OEO, or its main components (thymol and carvacrol), can destroy the cell membrane structure to exert antibacterial activity [3,7]. Furthermore, EOs has been widely used in poultry and pigs for many years. A supplement of OEO (containing thymol and carvacrol) could alleviate intestinal injuries in broiler chickens under *Clostridium perfringens* challenge [8], enhance the growth performance and intestinal health of broilers [9,10,11,12,13], inhibit the growth of pathogenic bacteria, and modulate the intestinal microbial composition of broilers [14,15]. 

The efficacy of EOs (with thymol, carvacrol, or menthol as active components) on improving laying performance and egg quality has been reported [16,17,18]. Cinnamaldehyde is the major compound of cinnamon essential oil, which can alter microbial composition [19]. It has been reported that cinnamaldehyde could enhance growth performance by improving intestinal histomorphology in broilers under a necrotic enteritis challenge [20]. Dietary supplementation of cinnamaldehyde could improve nutrient utilization and decrease total nitrogen excretion in broilers [21]. However, there are few investigations on the effects of OEO and cinnamaldehyde on the production performance, egg quality, and intestinal microbial community of laying hens in the late phase of production. 

In this study, we hypothesized that OEO may serve as an antibiotic alternative to positively alter microbial composition, subsequently leading to improvements in production performance, egg quality, and fatty acid composition of yolks. Therefore, the purpose of this study was to investigate the effects of dietary oregano essential oil supplementation on laying performance, egg quality, fatty acid composition in yolk, and microbial community in the cecum of laying hens in the late phase of production.

## 2. Material and Methods

### 2.1. Animals and Experimental Design

A total of 350 58-week-old Jing Tint Six laying hens were randomly divided into 5 groups with 5 replicates of 14 birds. The control group (CK) received a basal diet. The positive control group (AGP) received a basal diet supplemented with 5 mg/kg flavomycin. The three treatment groups (EO1, EO2, and EO3) received a basal diet and 20 mg/kg cinnamaldehyde supplemented with 100, 200, and 300 mg/kg oregano essential oil, respectively. The trial lasted 8 weeks and the birds were provided by Gu’an Songhe Poultry Breeding Co., Ltd. (Hebei China). An OEO product (containing 8% oregano essential oil) was obtained from a commercial supply (Ropapharm International B.V. Zaandam Netherlands) and its major active ingredients were carvacrol and thymol, with wheat flour and silicon dioxide as carriers. The actual concentrations of OEO in the EO1, EO2, and EO3 diets were 8, 16, and 24 mg/kg, respectively. The concentrations of carvacrol and thymol in the EO were ≥3.6% and ≥0.1%, respectively. The cinnamaldehyde was purchased from Shanghai Aladdin Biochemical Technology Co., Ltd. (Shanghai, China). All birds were housed in four-tier battery cages in an environmentally controlled house with the temperature maintained at approximately 23 °C and humidity maintained at 30–48%. All birds were provided ad libitum access to fresh water and mash feed. The basal diet (Table 1) was formulated according to NRC (1994) recommendations. All birds remained in good health and no medical intervention was applied to any birds during the feeding period.

### 2.2. Sample Collection 

Three eggs per replicate were randomly collected for egg quality determination on days 28 and 56. Two eggs in each replicate were randomly sampled on day 56 to separate the egg yolk from egg whites. The egg yolk was freeze-dried for fatty acid determination. On day 56, 1 bird per replicate (5 per treatment) was slaughtered by cervical dislocation after fasting for 12 h. Cecal contents were collected and kept frozen at −80 °C for assessment of microbial composition.

### 2.3. Laying Performance and Egg Quality

During the experimental period, egg production, broken-egg production, and egg weight were recorded daily by replicate, and feed consumption for each replicate was weighed every week. The feed conversion ratio (FCR) was calculated as grams of feed consumed/egg weight for each replicate. Egg length (mm) and width (mm) were recorded for shape index calculation (shape index = length/width). The breaking strength and thickness of eggshells were measured by an Egg Force Reader (EFR-01, Israel Orka Food Technology Ltd., Bountiful, UT, USA) and Egg Shell Thickness Gauge (ESTG-1, Israel Orka Food Technology Ltd., Bountiful, UT, USA), respectively. Egg weights, Haugh unit values, and albumen heights were measured by an Egg Analyzer (EA-01, Israel Orka Food Technology Ltd., Bountiful, UT, USA). 

### 2.4. Analysis of Ileal Microbiota

Total genome DNA from samples was extracted using the CTAB method. DNA concentration and purity was monitored on 1% agarose gels. According to the concentration, DNA was diluted to 1 ng/µL using sterile water. The V3-V4 region of the 16S rRNA gene was amplified using the primer pair 341F/806R (5′-CCTAYGGGRBGCASCAG-3′ and 5′-GGACTACHVGGGTWTCTAAT-3′). Sequencing libraries were generated using an Illumina TruSeq DNA PCR-Free Library Preparation Kit (Illumina, San Diego, CA, USA) following the manufacturer’s recommendations, and index codes were added. The library quality was assessed on the Qubit@ 2.0 Fluorometer (Thermo Scientific, Waltham, MA, USA) and Agilent Bioanalyzer 2100 system. The library was sequenced on an Illumina NovaSeq platform and 250 bp paired-end reads were generated. Sequencing and bioinformatics analysis were performed by Novogene Bioinformatics Technology Co. (Beijing, China).

### 2.5. Statistical Analysis 

Data were analyzed by a one-way Analysis of Variance (ANOVA) procedure and differences were examined using LSD’s Multiple Range Test by IBM SPSS Statistics 23. The differences in the relative abundances of bacteria between groups were assessed using Wilcoxon rank tests. Data were presented as mean ± SEM. Significant differences were defined as *p* < 0.05, and tendency was defined as 0.05 ≤ *p* < 0.10.

## 3. Results

### 3.1. Laying Performance and Egg Quality 

Dietary OEO supplementation had no significant influences (*p* > 0.05) on average daily feed intake (ADFI), or on broken-egg production of laying hens during weeks 1–8 of the experiment (Table 2). However, EO2 increased (*p* < 0.05) egg production of laying hens in comparison with the control in weeks 5–8 and 1–8. Average egg weight in group EO2 was lower (*p* < 0.05) than that in the AGP in weeks 1–4, but it was higher (*p* < 0.05) than that in the control during weeks 5–8 and 1–8. The EO2 group decreased (*p* < 0.05) the FCR when compared to groups EO1 and EO3. With respect to egg quality, there were no significant effects (*p* > 0.05) of dietary OEO or cinnamaldehyde supplementation on eggshell strength, shape index, albumen height, or Haugh unit at the end of weeks 4 or 8 (Table 3). Eggshell thickness in EO2 and EO3 was lower (*p* < 0.05) than that in the control and AGP in week 4. 

### 3.2. Yolk Fatty Acids Composition

Fatty acid composition and content of yolks at week 8 are shown in Table 4. The concentration of butyric acid (C4:0) in EO3 was significantly increased compared to the control and AGP groups. The concentration of lauric acid (C12:0) in EO1, EO2, and EO3 was significantly increased compared to the control group. The concentration of pentadecanoic acid (C15:0) in EO2 and EO3 was significantly lower than that in the control and AGP groups. The concentration of stearic acid (C18:0) in EO1 and EO3 was significantly increased compared to the control group. Yolk exhibited greater concentrations of oleic acid (C18:1, cis(n-9)) in EO2 and EO3, and greater concentrations of eicosenoic (C20:1) and eicosapentaeonic (C20:5) acids in EO1, EO2, and EO3, as compared to the control group (*p* < 0.05). Additionally, the concentration of MUFA in EO2 and EO3 was higher (*p* < 0.05) than that in the control and AGP groups at week 8.

### 3.3. Cecal Microbial Profile

As shown in Table 5, there was a tendency for Chao1 and ACE to increase in the EO2 group. The Venn analysis of ASVs identified 571, 432, and 971 unique ASVs in the control, AGP, and EO2 groups, respectively. These three treatments shared 895 ASVs among their cecum microbiota (Figure 1A). The PCA results suggested that there was a clear difference between the bacterial communities of the control group and EO2, while the separation between the control group and AGP could hardly be detected (Figure 1B). 

The dominant phyla in the three treatment groups were Bacteroidota and Firmicutes, together accounting for more than 75% of all phyla (Figure 2A). Laying hens from group EO2 had a higher abundance of Bacteroidota and a lower abundance of Firmicutes compared to the AGP group. At the class level, the dominant classes were Bacteroidia, Clostridia, and Fusobacteriia, which collectively accounted for more than 73% of the total sequences (Figure 2B). At the family level (Figure 2C), the cecal microbiota were dominated by Bacteroidaceae in all groups.

At the phylum level, the abundance of Desulfobacterota in the control group was higher (*p* < 0.05) than that in AGP and EO2, and EO2 had a higher (*p* < 0.05) abundance of Actinobacteriota than the control and AGP groups (Table 6). At the class level, the control had a higher (*p* < 0.05) abundance of Desulfovibrionia than AGP and EO2. At the family level, AGP had a higher (*p* < 0.05) abundance of Lachnospiraceae than the control and EO2, and the control had a higher (*p* < 0.05) abundance of Desulfovibrionaceae than AGP and EO2. At the genus level, the control had a higher (*p* < 0.05) abundance of *Desulfovibri* than AGP and EO2.

### 3.4. Correlation between Cecal Microbiota and Laying Performance

As shown in Figure 3, the egg production was positively correlated with the abundances of *Sutterella* and *Anaerofilum*, but it was negative correlated with *Lactobacillus*, *Methanobrevibacter*, *Desulfovibrio*, and *Rikenellaceae_RC9_gut_group* abundances. FCR was positively correlated with *Fusobacterium*, *Fournierella*, *Sutterella*, and *Anaerofilum*, while showing a negative correlation with *Bacteroides*. There was a positive correlation between ADFI and *Anaerofilum*, *Sutterella*, and *Fournierella*, and a negative correlation between ADFI and *Lactobacillus*, *Methanobrevibacter*, *Rikenellaceae_RC9_gut_group*, *Desulfovibrio*, and *Bacteroides*.

## 4. Discussion

Essential oils and their active ingredients have been extensively studied for their growth-promoting properties, which make them a potential alternative to AGPs [22]. OEO can increase the villus height and decrease the crypt depth, promoting the absorption of nutrients in the intestine [23]. In the present study, egg production was improved in the EO2 treatments, but there was no significant effect on FCR in groups EO1, EO2, and EO3 when compared to the control and AGP groups. In line with our study, several previous studies had reported that OEO improved egg production in laying hens [24,25]. This beneficial effect could be attributed to the active ingredients (i.e., thymol and carvacrol) in OEO and cinnamaldehyde, which have been proven to have antibacterial and anti-inflammatory properties, and to improve intestinal health and nutrient utilization [21,26]. In contrast, other studies had shown that OEO (or its main compounds) did not significantly improve laying performance of laying hens [27,28]. The inconsistent effects of OEO on production performance may be caused by the composition and supplemental levels of OEO, the type of laying hen, the feeding phase, and the environmental conditions. It is noteworthy that the FCR of the EO2 treatment group was lower (*p* < 0.05) than that of the EO3 group, which may be due to the negative effects of high concentrations of essential oils on intestinal epithelial cells and intestinal probiotics [29,30].

Eggshell thickness is an important indicator of egg quality and is critical for egg transportation and storage [31]. During eggshell calcification, the availability of intestinal calcium is vital as it plays a key role in providing sufficient amounts of calcium to satisfy shell quality requirements [32]. It has been reported that dietary essential oils could significantly increase eggshell thickness [24,33]. But in our study, the eggshell thickness of EO1 and EO2 was lower (*p* < 0.05) than that of the control and AGP at week 4, although the difference disappeared at week 8, which may be due to altered flora structure [34].

The lipid composition of eggs has received attention due to the relationship between dietary lipids and the development of coronary heart disease [35]. MUFAs have a significant impact on improving cardiovascular and cerebrovascular health and reducing oxidative stress damage in the body [36]. A MUFA-rich diet can improve insulin sensitivity, and positively affects blood lipids, systemic inflammatory response, and endothelial dysfunction [37]. Some countries have even advocated that MUFAs should become healthy substitutes in the daily diet [38]. Previous research has reported that dietary EOs and rosemary extract supplementation in the diet of laying hens did not significantly influence the fatty acid composition of their egg yolks [24,39]. Another study found that dietary bergamot oils could significantly increase the proportion of DHA and the n-3 PUFA proportion of egg yolk [40]. In the present study, the content of MUFAs in groups EO2 and EO3 was higher (*p* < 0.05) than that of the control and AGP treatments, which may be due to OEO and cinnamaldehyde affecting lipid metabolism in the serum and liver of laying hens [40,41].

Intestinal flora is involved in food digestion and metabolism, body growth and development, and immune suppression and activation. As one of the three major components of the intestinal barrier, intestinal flora is a direct factor in whether pathogenic bacteria can invade [42,43,44]. OEO can cause irreversible damage to bacteria cell membranes and can cause leakage of biological macromolecules. [45] The present study was performed to better understand the effect of OEO and cinnamaldehyde on microbiota. Since the beneficial effects of OEO on production performance were mainly observed in the EO2 group, the modulatory roles of OEO on cecal microbial composition were assessed in the control, AGP, and EO2 groups. ACE and Chao1 indices were used to estimate species richness, while Shannon and Simpson indices were used to estimate species diversity [46]. The Chao1 of EO2 was higher (*p* < 0.05) than that of the AGP group, indicating that the combination of OEO and cinnamaldehyde increased the abundance of cecal flora. According to the beta-diversity results of the three treatments, significant clustering was observed, indicating that the addition of OEO and cinnamaldehyde altered the cecal microbial community structure. Further analysis of the changes in microbiota composition and the specific taxa present upon addition of OEO and cinnamaldehyde was then performed. At the phylum level, Bacteroidota and Firmicutes, as dominant flora, accounted for more than 75% of the total microbial community. Firmicutes and Bacteroidetes are involved in the metabolism of nutrients in the body (such as enzymes encoding polysaccharide decomposition, involved in sugar metabolism), and are cooperatively involved in amino acid metabolism [47,48]. Actinobacteriota is reported to improve feed utilization by producing extracellular enzymes [49], and to decompose undigested components in feed through secreting endogenous enzymes [50]. In addition, Actinobacteriota is helpful for the maintenance of overall microbial structure. Due to the production of bacteriocins and the ability to convert feed into fermentable microbial biomass [51], Actinobacteriota is considered to be a key group that regulates the function of gut microbiota. In the present study, the abundance of Actinobacteriota in group EO2 was higher (*p* < 0.05) than that of the control and AGP, indicating that dietary supplementation OEO and cinnamaldehyde may increase the abundance of beneficial bacteria to promote the production performance and feed utilization of the late-stage laying hens. Short chain fatty acids (SCFAs) play important roles in the gut, such as inflammation reduction, cancer prevention, clearance of drug-resistant pathogenic bacteria, and regulation of gene expression [52,53,54]. Studies have shown that Lachnospiraceae can promote SCFAs production [55]. At the family level, the abundance of Lachnospiraceae in AGP was higher (*p* < 0.05) than that of the control and EO2, suggesting that AGP might promote production performance by increasing SCFA-producing bacteria, but the specific mechanism of action needs to be studied in depth. At the genus level, the abundance of *Desulfovibrio* in AGP and EO2 was lower (*p* < 0.05) than that of the control. *Desulfovibrio* are considered to be harmful bacteria, because *Desulfovibrio* acts as a hydrogen sink in the cecal ecosystem of birds, and hydrogen inhibits the production of short-chain fatty acids [56,57]. Therefore, *Desulfovibrio* was highly negatively correlated with laying hens’ egg production and ADFI in our study. *Lactobacilli* is considered a beneficial bacteria due to its tendency to reduce colonization by enteric pathogens through competitive exclusion, antagonistic activity, and production of bacteriocins [58]. Interestingly, the results of this study have shown that *Lactobacillus* was negatively correlated with laying hens’ egg production and ADFI, which may be related to *Lactobacillus* deconjugating bile acids, affecting lipid metabolism and energy utilization [59,60]. 

## 5. Conclusions 

Based on the results above, we conclude that dietary supplementation with OEO and cinnamaldehyde improved performance and decreased the feed conversion ratio of late-phase laying hens by the selective modulations of cecum microbial communities. These findings may provide useful information for developing an effective and safe alternative to AGP in the poultry industry. 

## Figures and Tables

**Figure 1 animals-12-03007-f001:**
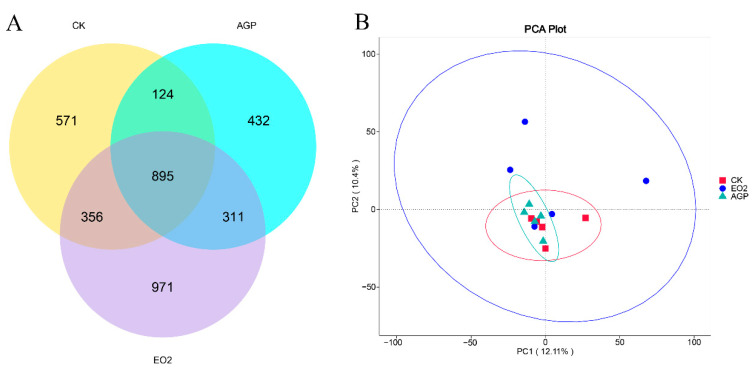
Venn diagram of ASV (**A**) and PCA (**B**) of cecum microbiota in the three groups. Control, hens received a basal diet; AGP, hens received a basal diet supplemented with 5 mg/kg flavomycin; EO2, hens received a basal diet supplemented with 200 mg/kg oregano essential oil and 20 mg/kg cinnamaldehyde.

**Figure 2 animals-12-03007-f002:**
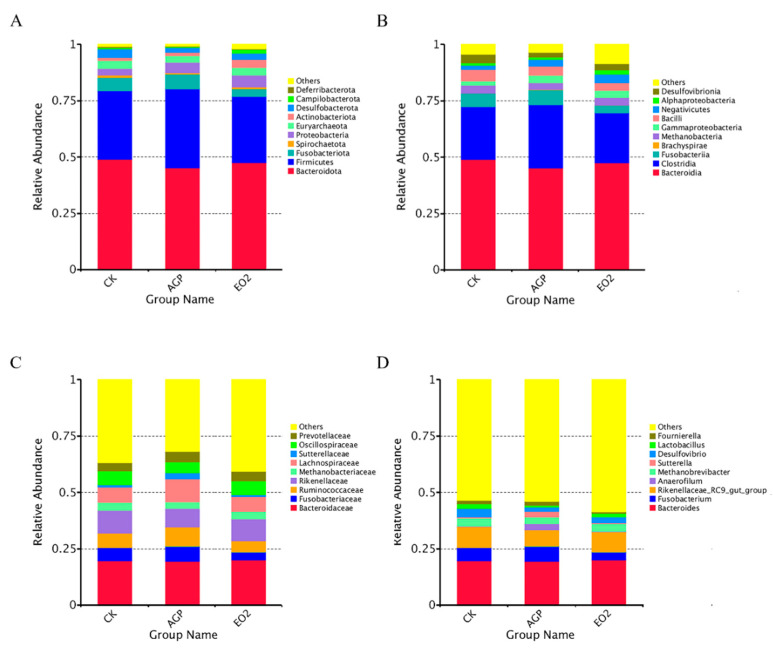
Microbial composition in the cecum of laying hens at phylum level (**A**), class level (**B**), family level (**C**), and genus level (**D**). Control, hens received a basal diet; AGP, hens received a basal diet supplemented with 5 mg/kg flavomycin; EO2, hens received a basal diet supplemented with 200 mg/kg oregano essential oil and 20 mg/kg cinnamaldehyde.

**Figure 3 animals-12-03007-f003:**
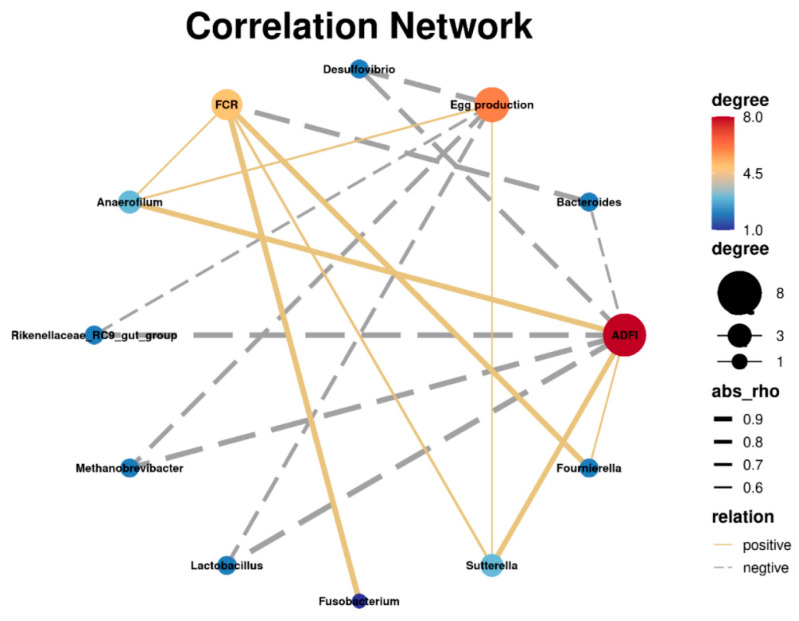
Pearson’s correlation analysis between the abundances of cecal microbiota (at genus level) and egg production. FCR, feed conversion ratio; ADFI, average daily feed intake.

**Table 1 animals-12-03007-t001:** Ingredient and nutrient levels of the experimental diets (%, as-fed basis).

Item	
Wheat	72.00
Soybean meal (44% CP)	14.50
Soybean oil	1.00
Limestone	10.30
Premix ^1^	2.20
Total	100.00
Calculated nutrient levels	
Crude protein	16.06
Calcium	3.86
Metabolizable energy, MJ/kg	10.96
Available phosphorus	0.32
SID lysine	0.66
SID methionine	0.37
SID threonine	0.52
SID tryptophan	0.19

^1^ Premix provided the following per kg of the diet: vitamin A, 443,040 IU; vitamin D3, 99,968 IU; vitamin E, 1136 IU; vitamin K_3_, 113.6 mg; vitamin B_1_, 113.6 mg; vitamin B_2_, 284.0 mg; vitamin B_6_, 170.4 mg; vitamin B_12_, 0.9 mg; D-biotin, 13.6 mg; D-pantothenic acid, 511.2 mg; nicotinamide, 1818 mg; Choline chloride, 13,638 mg; Fe, 3636 mg; Cu, 681.0 mg; Mn, 5454 mg; Zn, 4090 mg; L-Lysine, 7000 mg; DL-Methionine, 60,000 mg; NaCl, 3000 mg.

**Table 2 animals-12-03007-t002:** Effects of dietary supplementation with oregano essential oil on laying performance of laying hens ^1^.

Items	Treatments ^2^					SEM	*p*-Value
	Control	AGP	EO1	EO2	EO3		
Egg production, %							
Weeks 1–4	90.13 ^a^	91.32 ^a^	85.36 ^b^	91.42 ^a^	86.82 ^b^	0.89	0.001
Weeks 5–8	87.46 ^b^	92.38 ^a^	85.71 ^b^	91.95 ^a^	86.50 ^b^	0.90	0.001
Weeks 1–8	88.77 ^b^	91.86 ^a^	85.48 ^c^	91.64 ^a^	86.67 ^c^	0.67	0.001
Average egg weight, g							
Weeks 1–4	58.68 ^b^	59.46 ^a^	59.62 ^a^	58.97 ^b^	59.47 ^a^	0.19	0.001
Weeks 5–8	58.98 ^c^	59.65 ^ab^	59.96 ^a^	59.34 ^b^	59.93 ^a^	0.18	0.001
Weeks 1–8	58.81 ^c^	59.56 ^a^	59.795 ^a^	59.18 ^b^	59.70 ^a^	0.13	0.001
ADFI, g/hen per day							
Weeks 1–4	115.34	120.34	115.96	116.93	118.89	3.70	0.649
Weeks 5–8	109.61	116.34	109.63	111.03	114.81	3.95	0.389
Weeks 1–8	112.47	118.32	112.79	113.98	116.85	2.87	0.193
FCR, g/g							
Weeks 1–4	2.18	2.22	2.28	2.16	2.30	0.06	0.117
Weeks 5–8	2.25	2.24	2.27	2.15	2.32	0.08	0.349
Weeks 1–8	2.22 ^ab^	2.23 ^ab^	2.28 ^a^	2.16 ^b^	2.31 ^a^	0.05	0.026
Broken-egg production							
Weeks 1–4	1.57	3.20	3.61	1.78	2.21	0.78	0.076
Weeks 5–8	3.60	4.91	3.09	3.34	5.21	0.85	0.082
Weeks 1–8	2.58	4.05	3.35	2.60	3.72	0.75	0.203

^1^*n* = 5 replicates per treatment. ^2^ Control, a basal diet; AGP, a basal diet + 5 mg/kg flavomycin; EO1, a basal diet + 100 mg/kg oregano essential oil + 20 mg/kg cinnamaldehyde; EO2, a basal diet + 200 mg/kg oregano essential oil + 20 mg/kg cinnamaldehyde; and EO3, a basal diet + 300 mg/kg oregano essential oil + 20 mg/kg cinnamaldehyde. ADFI, average daily feed intake; FCR, feed conversion ratio. ^a–c^ Values within a row with no common superscripts differ significantly (*p* < 0.05).

**Table 3 animals-12-03007-t003:** Effects of dietary supplementation with oregano essential oil on egg quality of laying hens ^1^.

Items	Treatments ^2^					SEM	*p*-Value
	Control	AGP	EO1	EO2	EO3		
Eggshell thickness, 10^−2^ mm							
Week 4	28.78 ^a^	28.56 ^a^	28.24 ^ab^	27.91 ^b^	27.87 ^b^	0.27	0.005
Week 8	28.73	28.89	29.02	29.42	29.20	0.47	0.643
Eggshell strength, kg							
Week 4	3.55	3.45	3.45	3.46	3.57	0.24	0.971
Week 8	3.42	3.41	3.45	361	3.55	0.26	0.918
Shape index							
Week 4	1.35	1.34	1.34	1.36	1.38	0.02	0.080
Week 8	1.34	1.35	1.34	1.34	1.37	0.02	0.160
Albumen height, mm							
Week 4	6.53	6.44	6.14	6.40	5.76	0.43	0.393
Week 8	5.99	6.05	5.98	5.96	5.90	0.44	0.998
Haugh unit							
Week 4	79.53	78.96	77.83	78.77	73.65	3.51	0.462
Week 8	75.70	75.81	76.21	76.70	74.28	3.66	0.974

^1^*n* = 5 replicates per treatment. ^2^ Control, a basal diet; AGP, a basal diet + 5 mg/kg flavomycin; EO1, a basal diet + 100 mg/kg oregano essential oil + 20 mg/kg cinnamaldehyde; EO2, a basal diet + 200 mg/kg oregano essential oil + 20 mg/kg cinnamaldehyde; and EO3, a basal diet + 300 mg/kg oregano essential oil + 20 mg/kg cinnamaldehyde. ^a–c^ Values within a row with no common superscripts differ significantly (*p* < 0.05).

**Table 4 animals-12-03007-t004:** Effects of dietary supplementation with oregano essential oil on fatty acid composition of egg yolk. (g/100 g) ^1^.

Fatty Acid	Treatments ^2^					SEM	*p*-Value
	Control	AGP	EO1	EO2	EO3		
C4:0	0.002 ^b^	0.003 ^b^	0.005 ^b^	0.005 ^b^	0.008 ^a^	0.001	0.005
C12:0	0.005 ^b^	0.006 ^a^	0.006 ^a^	0.006 ^a^	0.006 ^a^	0.000	0.003
C14:0	0.277	0.273	0.269	0.267	0.271	0.009	0.814
C15:0	0.044 ^a^	0.043 ^a^	0.041 ^ab^	0.037 ^bc^	0.034 ^c^	0.003	0.006
C16:0	18.656	18.581	19.411	19.050	19.220	0.554	0.520
C18:0	6.136 ^b^	6.565 ^a^	6.748 ^a^	6.499 ^b^	6.624 ^a^	0.175	0.026
C21:0	0.047	0.057	0.060	0.059	0.057	0.004	0.056
C24:0	0.097	0.092	0.091	0.090	0.098	0.007	0.716
C16:1	2.660	2.818	2.664	2.799	2.753	0.186	0.866
C18:1, cis(n-9)	15.191 ^c^	15.245 ^bc^	15.490 ^bc^	16.069 ^ab^	16.452 ^a^	0.379	0.012
C20:1 n9	0.131 ^b^	0.148 ^ab^	0.156 ^a^	0.158 ^a^	0.153 ^a^	0.008	0.034
C18:2, cis(n-6)	9.518	9.505	9.336	9.157	9.573	0.472	0.899
C18:3 n-3	0.360	0.368	0.361	0.333	0.349	0.025	0.675
C20:2	0.138	0.153	0.159	0.148	0.142	0.007	0.059
C20:3 n-6	0.149	0.154	0.155	0.143	0.139	0.006	0.057
C20:4 n-6	1.397	1.430	1.420	1.382	1.460	0.050	0.578
C20:5	0.010 ^b^	0.013 ^a^	0.014 ^a^	0.013 ^a^	0.014 ^a^	0.001	0.001
C22:6	0.701	0.731	0.713	0.715	0.780	0.037	0.292
SFA	25.392	26.309	26.778	26.013	26.319	0.599	0.204
UFA	30.260	30.565	30.468	30.917	31.816	0.529	0.061
MUFA	17.983 ^c^	18.212 ^c^	18.311 ^bc^	19.025 ^ab^	19.358 ^a^	0.349	0.003
PUFA	12.276	12.355	12.158	11.891	12.457	0.552	0.867
n-3 PUFA	1.062	1.100	1.074	1.048	1.129	0.056	0.627
n-6 PUFA	11.064	11.089	10.911	10.682	11.172	0.501	0.876
n-3/n-6	0.096	0.099	0.099	0.098	0.101	0.003	0.605
UFA/SFA	1.197	1.196	1.144	1.189	1.210	0.028	0.221
EFA	11.276	11.303	11.117	10.872	11.383	0.522	0.874

^1^*n* = 5 replicates per treatment. ^2^ Control, a basal diet; AGP, a basal diet + 5 mg/kg flavomycin; EO1, a basal diet + 100 mg/kg oregano essential oil + 20 mg/kg cinnamaldehyde; EO2, a basal diet + 200 mg/kg oregano essential oil + 20 mg/kg cinnamaldehyde; and EO3, a basal diet + 300 mg/kg oregano essential oil + 20 mg/kg cinnamaldehyde. SFA, saturated fatty acid; UFA, unsaturated fatty acid; MUFA, monounsaturated fatty acid; PUFA, polyunsaturated fatty acid; EFA, essential fatty acids; n-3 PUFA= C18:3 n-3 + C22:6; n-6 PUFA= C18:2, cis(n-6) + C20:3 n-6+ C20:4 n-6. ^a–c^ Values within a row with no common superscripts differ significantly (*p* < 0.05).

**Table 5 animals-12-03007-t005:** Effect of dietary supplementation with oregano essential oil on alpha diversity in cecum ^1^.

Items	Treatments ^2^			SEM	*p*-Value
	Control	AGP	EO2		
Shannon	7.94	7.84	8.18	0.32	0.579
Simpson	0.99	0.98	0.99	0.01	0.627
Chao1	855.41	790.28	1006.06	79.45	0.050
ACE	850.20	786.00	996.60	78.94	0.055

^1^*n* = 5 replicates per treatment. ^2^ Control, a basal diet; AGP, a basal diet + 5 mg/kg flavomycin; EO2, a basal diet + 200 mg/kg oregano essential oil + 20 mg/kg cinnamaldehyde.

**Table 6 animals-12-03007-t006:** Differences of bacterial distribution in cecal digesta between the control, AGP, and EO2 groups ^1^.

	Treatments ^2^			SEM	*p*-Value
Items, %	Control ^2^	AGP	EO2		
Phylum					
Desulfobacterota	3.72 ^a^	2.04 ^b^	2.66 ^b^	0.45	0.010
Actinobacteriota	1.34 ^b^	1.51 ^b^	3.88 ^a^	0.91	0.016
Class					
Desulfovibrionia	3.69 ^a^	2.02 ^b^	2.61 ^b^	0.44	0.008
Family					
Lachnospiraceae	6.79 ^b^	10.16 ^a^	6.69 ^b^	0.85	0.002
Desulfovibrionaceae	3.69 ^a^	2.02 ^b^	2.61 ^b^	0.44	0.008
Genus					
*Desulfovibrio*	3.63 ^a^	2.00 ^b^	2.56 ^b^	0.44	0.010

^1^*n* = 5 replicates per treatment. ^2^ Control, a basal diet; AGP, a basal diet + 5 mg/kg flavomycin; EO2, a basal diet + 200 mg/kg oregano essential oil + 20 mg/kg cinnamaldehyde. ^a,b^ Values within a row with no common superscripts differ significantly (*p* < 0.05).

## Data Availability

All data are contained within the article.

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
