# Peer review of "Dietary Oregano Essential Oil Supplementation Influences Production Performance and Gut Microbiota in Late-Phase Laying Hens Fed Wheat-Based Diets"

_animals, 2022, doi:10.3390/ani12213007_

Round 1

Reviewer 1 Report

All my comments are in the attached PDF file

However, authors should change the focus of the paper because they did not vary the cinnamaldehyde, thus it is impossible specify its effect on the evaluated parameters. 

There are some mistakes in the section Results. 

Author Response

Dear Reviewer

Good day. Thank you very much for your kind consideration with our submitted article and offering us the further opportunity to submit the revised manuscript. The amendments are highlighted in red in the revised manuscript. We have revised our manuscript for language and grammar checked by a native English speaker working in our university. We do thanks to your critical evaluation to make the manuscript more effective for review process in Animals Journal.

Thank you and best regards.

Point 1: All my comments are in the attached PDF file

Response 1: Thanks for your suggestion. I have amended the comments in the revised manuscript.

Point 2: However, authors should change the focus of the paper because they did not vary the cinnamaldehyde, thus it is impossible specify its effect on the evaluated parameters.

Response 2: Thanks for your available comment. I have made some changes that some cinnamaldehyde content has been removed.

Point 3: There are some mistakes in the section Results.

Response 3: Thanks for your advice. I have modified the mistakes in the results section. See line 171-180.

Reviewer 2 Report

This is an interesting paper reporting novel findings useful for poultry nutrition and poultry producers.

The full-text has been well organized and each section is clear and understable for readers.

The results have been properly discussed by using updated literature.

As specific comments, I suggest to:

- In the Introduction add a couple of additional recently published references;

- In Table 1, report the ingredients based on their inclusion level, and add the crude protein % of soybean meal used;

- Please, spell all the acronyms used in Tables as footnote...this also for acronyms in the full-text;

- Report the Conclusions as new section.

So, I recommend the acceptance of the paper after minor revisions.

Author Response

Dear Reviewer

Good day. Thank you very much for your kind consideration with our submitted article and offering us the further opportunity to submit the revised manuscript. The amendments are highlighted in red in the revised manuscript. We have revised our manuscript for language and grammar checked by a native English speaker working in our university. We do thanks to your critical evaluation to make the manuscript more effective for review process in Animals Journal.

Thank you and best regards.

Point 1: This is an interesting paper reporting novel findings useful for poultry nutrition and poultry producers.

Response 1: Thanks for your available comment.

Point 2: The full-text has been well organized and each section is clear and understable for readers.

Response 2: Thanks for your available comment.

Point 3: The results have been properly discussed by using updated literature.

Response 3: Thanks for your suggestion. I have modified it according to your request. See line 247-248, and 287-288.

Point 4: In the Introduction add a couple of additional recently published references;

Response 4: Thanks for your suggestion. I have cited the new references in the introduction. See line 365-367, 382-387, and 414-416.

Point 5: In Table 1, report the ingredients based on their inclusion level, and add the crude protein % of soybean meal used;

Response 5: Thanks for your available comment. I have modified it according to your request. See line 102.

Point 6: Please, spell all the acronyms used in Tables as footnote...this also for acronyms in the full-text;

Response 6: Thanks for your advice. I have modified it according to your request. See line 162.

Point 7: Report the Conclusions as new section.

Response 7: Thanks for your advice. I have made the conclusion a new paragraph. See line 331.

Reviewer 3 Report

There is descripancy of abbreviations throughout the paper. They must be corrected. from the conclusion it seems that the authors are not confident much to conclude. what is the reason of combining these two agents in nature and doses??

add the following reference: Tropical animal health and production 52 (5), 2499-2504

authors should put stress on the mechanism of action of these herbal agents while discussing the studied parameters. 

conclusion should be brief and to the point based on the results obtained. 

Author Response

Dear Reviewer

Good day. Thank you very much for your kind consideration with our submitted article and offering us the further opportunity to submit the revised manuscript. The amendments are highlighted in red in the revised manuscript. We have revised our manuscript for language and grammar checked by a native English speaker working in our university. We do thanks to your critical evaluation to make the manuscript more effective for review process in Animals Journal.

Thank you and best regards.

Point 1: There is descripancy of abbreviations throughout the paper. They must be corrected. from the conclusion it seems that the authors are not confident much to conclude. what is the reason of combining these two agents in nature and doses??

Response 1: Thanks for your suggestion. I have modified it according to your request. Both oregano essential oil and cinnamaldehyde have strong antimicrobial properties, so we combined them together. The dose was determined based on the references we reviewed.

Point 2: add the following reference: Tropical animal health and production 52 (5), 2499-2504

Response 2: Thanks for your available comment. I have added the reference. See line 382-384.

Point 3: authors should put stress on the mechanism of action of these herbal agents while discussing the studied parameters.

Response 3: Thanks for your advice. I have modified it according to your request. See line 247-248, and 287-288.

Point 4: conclusion should be brief and to the point based on the results obtained.

Response 4: Thanks for your advice. I have modified conclusion according to your request. See line 332-337.

Round 2

Reviewer 1 Report

Authors really took into account all the comments and suggestions.